# Bibliometric and Density Visualisation Mapping Analysis of Domestic Violence in Australia Research Output 1984–2019

**DOI:** 10.3390/ijerph19084837

**Published:** 2022-04-15

**Authors:** Chloe Charlton, Ravishankar Ram Mani, Sasikala Chinnappan, Ashok Kumar Balaraman, Thangavel Muthusamy, Chitraabaanu Paranjothy, Deepa Suresh, Sunil Krishnan, Kartik Lokhotiya, Gothandam Kodiveri Muthukaliannan, Siddhartha Baxi, Rama Jayaraj

**Affiliations:** 1Department of Veterans Affairs, Charles Darwin University, Darwin, NT 0810, Australia; chloe_charlton@hotmail.com; 2Department of Pharmaceutical Biology, Faculty of Pharmaceutical Sciences, UCSI University Kuala Lumpur (South Wing), Kuala Lumpur 56000, Malaysia; ravishankar@ucsiuniversity.edu.my (R.R.M.); sasikala@ucsiuniversity.edu.my (S.C.); ashokkumar@ucsiuniversity.edu.my (A.K.B.); 3Research and Development Wing, Sri Balaji Medical College and Hospital, Bharath Institute of Higher Education and Research, Chromepet, Chennai 600044, Tamil Nadu, India; thangavelmuthusamy.research@bharathuniv.ac.in; 4Department of Health, University of Essex, Leeds LS1 2RP, UK; cb.mrsram@gmail.com; 5Division of Endocrinology, Department of Internal Medicine, Mayo Clinic Florida, Jacksonville, FL 32224, USA; deepa.suresh@mayo.edu; 6Department of Radiation Oncology, Mayo Clinic Florida, 4500 San Pablo Road S, Jacksonville, FL 32224, USA; krishnan.sunil@mayo.edu; 7Vellore Institute of Technology (VIT), School of Biosciences and Technology, Vellore 632014, Tamil Nadu, India; kartik.lakhotiya2016@vitalum.ac.in (K.L.); gothandam@gmail.com (G.K.M.); 8Genesis Care Gold Coast Radiation Oncologist, John Flynn Hospital, Tugun, QLD 4224, Australia; siddhartha.baxi@genesiscare.com; 9Clinical Sciences, Northern Territory Institute of Research and Training, Darwin, NT 0909, Australia

**Keywords:** domestic violence, family violence, intimate partner violence, bibliometric analysis, VOSviewer

## Abstract

Domestic violence is highly prevalent in Australia and has serious and complex impacts. This study aimed to analyse research outputs on domestic violence in Australia from the period of 1984 to 2019. Articles relevant to domestic violence in Australia that met specified inclusion criteria were retrieved using the Scopus database. Bibliometric analysis of the output was conducted to examine trends in publications. A trend of an increase in publications relating to domestic violence in Australia over time was identified, with the majority published in institutions located in densely populated capital cities. Significant diversity was found in the subject matter of highly cited articles, reflecting the far-reaching impacts of domestic violence. The increase in social attention to domestic violence over time was reflected in an increase in publications. Future research would benefit from examining trends in the reporting of domestic violence, and analysing the effectiveness of interventions for perpetrators and victims.

## 1. Introduction

### 1.1. Definition and Epidemiology

Domestic violence refers to acts of violence that occur between people who are, or have had, an intimate relationship in a domestic setting [1]. It may be perpetrated by either men or women, however women are more likely to be victims of domestic violence [2,3,4]. Domestic violence may include emotional, psychological, sexual, physical, verbal, and economic abuse [3,4]. In 2013, the World Health Organisation [2] released a report concerning worldwide violence against women and found an estimated 30 per cent of women worldwide had experienced physical and sexual intimate partner violence [2].

### 1.2. Impacts in Australia

Analysis of homicide statistics in Australia between 2007 and 2014 indicated that the most likely cause of homicide for an Australian woman was at the hands of an intimate partner [3,4,5]. Burden of disease studies, which measure the combined impact of living with illness and injury and dying prematurely in a population, suggest domestic violence as one of the leading causes of death, illness, and injury in Australian women aged between 18 to 44, with a disease burden of 5.1 per cent. In Australia, exposure of children to domestic violence is recognised as a form of child abuse, and frequently leads to a myriad of psychological, behavioural, and health problems, including depression and anxiety, low self-esteem, antisocial behaviour, and aggression [3,4,6].

### 1.3. Domestic Violence in Indigenous Populations

A 2011 review examined patterns of domestic violence among Indigenous Australians and found violence was much more prevalent for this group than in any other cultural group within Australian society, to the extent that violence was regarded as an inevitable part of life (Steering Committee for the Review of Government Service Providers, 2011). Reports of domestic violence to police were found to be six times higher than that of other cultural groups in Australia, and the reported violence tended to be more severe [7,8]. Domestic violence among Indigenous Australians was also found to be more likely to occur in a public place and involve multiple family members [4,9,10,11,12].

However, Indigenous Australians were less likely to report domestic violence than non-Indigenous Australians, due to less anonymity in a small community and greater risk of repercussions, fear and distrust of police, cultural considerations such as shame and the importance of maintaining family, and a lack of awareness or access to support [4,10,13,14,15,16].

### 1.4. Bibliometric Profile

A bibliometric profile provides information about research trends over time to researchers, policy makers, and practitioners [17,18]. Sweileh and colleagues’ (2016) bibliometric profile of research about autism spectrum disorders is well written and articulates the findings in a clear and informative manner. As such, the above-mentioned paper was drawn on as an example of an appropriate structure, and factors to consider for analysis and layout [19]. Analysis of trends in the literature sheds light on the density of publications by location, institution, journal, and authors, and considers the impact of these publications through analysing factors including citations and *h*-index [17]. A bibliometric profile also provides information about sources of funding for research, frequently co-occurring search terms, and prolific authors and institutions contributing to the body of literature [18]. Providing a bibliometric profile of publications concerning domestic violence in Australia will illustrate trends in existing research, highlight gaps in existing research, and therefore provide direction as to priorities for future research [18].

In light of the prevalence, impact, and increasing profile of domestic violence worldwide, there has been a gradual increase in research in this area over time [3]. A review of the available literature suggested patterns in these publications had not been analysed. Franco, Lopez-Cepero and Rodrigues (2009) conducted a bibliometric review of worldwide publications concerning domestic violence, however, this article was only published in Spanish [20]. Jordan (2009) reviewed the available body of research regarding violence against women, specifically examining methodological issues and conducting a bibliometric analysis [21]. This review found that research in the area of violence against women increased in funding between 1995 and 2005, and observed a consistent growth in the literature between 1977 to 2007. The disciplines contributing the majority of publications were psychology, psychiatry, nursing, medicine, sociology, epidemiology, public health, social work, pharmacy, statistics, criminal justice, and education [21].

Frank, Coehlho and Boing (2010) published a further review in Portuguese that reviewed publications concerning intimate partner violence between 2003 to 2007. The authors attributed this increase to greater visibility of the theme of intimate partner violence in the scientific community [22]. Lastly, Wang and colleagues (2015) published a bibliometric analysis of worldwide literature on intimate partner violence over a ten-year period, from 2005 to 2014. This article appeared to analyse trends in research during this period, however, outcomes were unclear due to the publication being in Chinese [23]. It appeared no bibliometric analyses had been conducted concerning publications on domestic violence and family violence in Australia.

### 1.5. Rationale and Objectives

As identified, domestic and family violence is recognised as a prevalent, pervasive issue nationwide, with long-lasting detrimental and potentially devastating consequences for those involved. The incidence of domestic violence is high in Australia, as an estimated 34 per cent of women have been a victim of domestic violence, and this number is estimated to be higher for Indigenous women [3]. Significant impacts of domestic violence include detrimental effects on health, social, and financial domains for victims, perpetrators, and children [3]. The financial cost of domestic violence was estimated to be 8.1 billion dollars in the 2002–2003 period, which was expected to rise to 9.9 billion dollars in 2021–2022 if intervention did not occur [3]. The Australian Government has identified responding to domestic violence as a priority, in identifying incidents, supporting victims, and providing appropriate interventions for perpetrators [2,3,4].

As awareness regarding domestic violence has increased, research and publications on the topic have also increased, however trends in these publications in Australia are yet to be explored through a bibliometric review. The analysis of bibliometric characteristics of domestic violence research in Australia allows the consolidation of knowledge on the available Australian scientific research evidence. This in turn could help practitioners, policy makers, governments, justice departments, and researchers to better understand the research field and learn more about current research trends, their impact, and scientific collaboration.

Therefore, the aim of this study was to analyse the Australian research publications on domestic and family violence using a bibliographic and density visualisation mapping analysis of articles indexed in Scopus.

## 2. Methodology

The data for the bibliometric profile were collected and synthesised through the Scopus database. Scopus was selected due to the broad access to a significant number of resources [24]. Search terms input were “Australia” AND “domestic violence” OR “family violence” OR “intimate partner violence”. The keywords were selected based on popular terms used in research concerning domestic violence, and specified to an Australian population. Ethics approval was not required as the data were publicly available. The data were retrieved from Scopus on the 18 February 2019.

The selection criteria for articles included in the bibliometric analysis were as follows:

### 2.1. Inclusion Criteria

Domestic violence papers focused on Australian data.Documents in all languages were included.Documents with police records and hospital incidences related to domestic and family violence were considered.

### 2.2. Exclusion Criteria

Documents categorised as abstracts, dissertation reports, case reports, case series, or un-defined types of documents were excluded.Documents that contained some of the keywords but the major focus of the article was not related to domestic or family violence were excluded.

The collected data were transferred to an MS Office Excel sheet that permitted management of the information contained in the records to retrieve bibliometric indicators.

The main bibliometric indicators presented in this study included type and language of the published documents, countries in which the sources were published, most prolific institutions and authors, journals with the greatest number of publications on the topic, most frequently cited articles, and sources of funding for research. The majority of bibliometric indicators were presented in rank order, from 1 to 10, based on number of published documents, the total number of citations, and the *h*-index.

### 2.3. Scopus Publication Indicators

Scientific Journal Ranking (SJR) scores were obtained from the SCImago Journal and Country Ranking website and were considered in the assessment of journal quality. Source Normalised Impact per Paper (SNIP) and CiteScore was also considered in the ranking of journals and were sourced through Scopus [25]. The quantity and quality of publications per country, institution or author were assessed using the Hirsch index (*h*-index). The *h*-index assesses scientific output based on the number of citations per paper in contrast to other publications [26]. To provide a visual representation of some of the bibliometric information, density visualisation mappings were generated using VOSviewer software [27]. Density visualisation mapping was conducted on search terms and authors, with more frequent prolific authors and more frequently occurring terms presenting with dense colour clusters.

## 3. Results

### 3.1. Trends in Yearly Publications

Of the 673 documents retrieved, the earliest date of publication was 1984 with approximately 17% of documents published before 2004 (116). An overall trend of an increasing number of publications per year was observed, and this trend appears to be accelerating, with over half of the documents having been published between 2011 and 2018 (369) (Figure 1). The most prolific year of publications was 2018, with the greatest number of publications (74, 11%) and citations (1052, 12.72%) per year, indicating the continuation of the trend. Of note, however, was that the highest average citation per document was recorded in 2016, with an average citation of 26 per document, in contrast to 14 per document in 2018.

### 3.2. Type of Documents

The overwhelming majority of documents retrieved were articles, as illustrated by Table 1 (506, 75%).

### 3.3. Language

The overwhelming majority of documents retrieved were in English (669, 99%), which is consistent with the search terms and location of publication. One publication each was retrieved in French, German, Spanish, and Italian.

### 3.4. Most Frequently Used Search Terms

In mapping the frequency search terms network, 160 terms were found to occur at least 14 times. Figure 2 shows the density visualisation map of most frequently encountered search terms in retrieved publications on domestic violence in Australia. The term “domestic violence” (395) occurred more frequently than “family violence” (135) and “partner violence” (93), which is perhaps a reflection of a gradual shift in the language regarding domestic violence over time, with “family violence” and “partner violence” emerging in the literature in more recent years. The term “female” (319) occurred with greater frequency than “male” (187), and “human/s” (701) occurred with greater frequency than each of the sex-based terms (Figure 3). The top 10 most frequently used search terms are outlined in Table 2.

### 3.5. Countries of Publication

A total of 10 countries have contributed publications relating to domestic violence in Australia. Table 3 shows these countries ranked in order of the number of contributions. As would be expected, Australia contributed the overwhelming majority, having published 79% of all documents. The United States and the UK each contributed approximately 7%, with the remaining countries accounting for approximately 9.5%.

### 3.6. Most Publishing Authors

Professor Kelsey Hegarty from the University of Melbourne ranked first in the number of articles contributed to the field of domestic violence in Australia (26), with an *h*-index of 9 (Table 4). Professor Hegarty is also ranked first for the number of citations (624). All of the top 10 authors contributing to this area of research are from Australia (Table 5).

**Table 4 ijerph-19-04837-t004:** Top 10 active authors publishing on domestic violence in Australia.

Rank	Author	Number of Published Articles (%)	Total Citation	*h*-Index	Country
1	Hegarty, K.	26 (3.86)	624 (7.54)	9	Australia
2	Humphreys, C.	15 (2.23)	118 (1.43)	7	Australia
3	Taft, A.J.	10 (1.49)	254 (3.07)	8	Australia
4	Lawrence, J.M.	8 (1.19)	306 (3.70)	7	Australia
5	Raphael, B.	8 (1.19)	398 (4.81)	7	Australia
6	Loxton, D.	7 (1.04)	122 (1.47)	5	Australia
7	Roberts, G.L.	7 (1.04)	301 (3.64)	7	Australia
8	Diemer, K.	6 (0.89)	7 (0.08)	2	Australia
9	Douglas, H.	6 (0.89)	79 (0.96)	4	Australia
10	Quinlivan, J.A.	6 (0.89)	144 (1.72)	4	Australia

**Table 5 ijerph-19-04837-t005:** The top 10 cited articles concerning domestic violence in Australia.

Rank	Authors	Title	Year	Source Title	Country	Institution	Number of Citations
1	Flemming et al. [28]	The long-term impact of childhood sexual abuse in Australian women	1999	Child Abuse and Neglect	Australia	Australian National University	187
2	Guthrie et al. [29]	The menopausal transition: A 9-year prospective population-based study. The Melbourne Women’s Midlife Health Project	2004	Climacteric	Australia	The University of Melbourne	174
3	Rees et al. [30]	Lifetime prevalence of gender-based violence in women and the relationship with mental disorders and psychosocial function	2011	Journal of the American Medical Association	Australia	University of New South Wales	173
4	Devries et al. [31]	Intimate partner violence during pregnancy: Analysis of prevalence data from 19 countries	2010	Reproductive Health Matters	England	London School of Hygiene and Tropical Medicine	144
5	Hegarty et al. [32]	The Composite Abuse Scale: Further development and assessment of reliability and validity of a multidimensional partner abuse measure in clinical settings	2005	Violence and Victims	Australia	The University of Melbourne	138
6	Gilbert et al. [33]	Child maltreatment: Variation in trends and policies in six developed countries	2012	The Lancet	England	University College London	137
7	Mazza et al. [34]	Physical, sexual and emotional violence against women: A general practice-based prevalence study	1996	Medical Journal of Australia	Australia	Monash University	112
8	Roberts et al. [35]	The impact of domestic violence on women’s mental health	1998	Australian and New Zealand Journal of Public Health	Australia	University of Queensland	100
9	Martijn and Sharpe [36]	Pathways to youth homelessness	2006	Social Science and Medicine	Australia	The University of Sydney	99
10	Rosenman and Rodgers [37]	Childhood adversity in an Australian population	2004	Social Psychiatry and Psychiatric Epidemiology	Australia	Australian National University	98

### 3.7. Frequently Cited Articles

The top 10 cited articles regarding domestic violence in Australia are outlined in the table below. The most frequently cited article is titled “The long-term impact of childhood sexual abuse in Australian women”, which was published in 1999 and had been cited 187 times (Table 5). The top 10 articles reflect a diverse range of subject matter, including sexual abuse, menopausal transition, the relationship between domestic violence and psychological functioning, development of an abuse assessment scale, and youth homelessness. This diversity is reflective of the pervasive impact of domestic violence across numerous life domains and the breadth of research being conducted concerning domestic violence in Australia. Of the 673 publications retrieved, 520 were cited at least once (77%), and 127 were cited 20 times or more (19%). Of the 152 articles that had not been cited, 82 were published between 2016 and 2019, suggesting recency of publication was a factor.

### 3.8. Institutions

As expected, the top 10 institutions contributing to the field of research in domestic violence in Australia are all Australian institutions (listed in Table 6). The University of Melbourne has published 86 articles with 1644 citations, contributing approximately 13% of all published material. Six of the top ten contributing institutions produced the highest number of publications regarding domestic violence in Australia in 2018, with the University of Melbourne almost doubling (15) their previous highest number of publications per year on the topic (8 in 2008). A distinct trend was also observed of an increase in citations for each of the top 10 institutions over time. Of the total 160 contributing institutions, 20 recorded more than a total of 10 publications on the topic.

### 3.9. Journals

The top 10 journals contributing to the publication of materials relating to domestic violence in Australia are listed in Table 7. The *Medical Journal of Australia* had the highest number of articles and citations by a significant amount. *Child Abuse and Neglect* was found to have a higher number of publications and citations than many others higher in the list, however, due to publishing a significantly lower number of reviewed publications it was ranked number 10.

### 3.10. Funding Resources

Funding for the analysed publications in the area of domestic violence in Australia was identified to come from 81 varying sources. The top 10 sources of funding are identified in Table 8. The vast majority of funding behind publications in the area of domestic violence in Australia is from the Australian Research Council (ARC), followed by the National Health and Medical Research Council (NHMRC).

## 4. Analysis

The purpose of the current study was to analyse the bibliometric characteristics of domestic violence research in Australia and consolidate available knowledge. It was anticipated that this would support practitioners, policy makers, and researchers to better understand the research field and learn more about current research trends and their impact. A review of the available literature demonstrated that no specific studies had been published that consolidated the information on research on domestic violence in Australia in this way. Therefore, this study sought to address this gap and provide insight as to existing trends and potential future directions of research into domestic and family violence in Australia.

Key findings showed that the highest density of publications on domestic violence in Australia was in Melbourne, followed by Brisbane and Sydney. This suggested a correlation between population density and publication density, and therefore publications concerning domestic violence are not being produced in the areas most affected by domestic violence, namely more rural and remote areas. Indigenous women and women in rural and remote areas have been identified as at-risk groups for domestic violence [3,4]. A research focus on these groups is not reflected in the contributions from institutions in more regional areas, and states and territories with a higher density of Indigenous people such as the Northern Territory or Western Australia [12]. It is therefore recommended that future research explores this trend, and specifically identifies whether greater attention needs to be placed on research into domestic violence in rural and remote areas.

The top 10 ranked articles reflect the significant range in the subject matter of publications connected to domestic violence, including sexual abuse, menopausal transition, the relationship between domestic violence and psychological functioning, development of an abuse assessment scale, and youth homelessness. This finding emphasises the pervasiveness of domestic violence across different populations and its far-reaching impact. It is recommended that additional data analysis be conducted to develop a clearer understanding of the subject matter of the body of literature, and identify any existing gaps in the literature.

The Australian Government’s increased awareness and prioritisation of domestic violence were reflected in the sources of funding of research, with the majority of funding being provided by the ARC. Further analysis of the specific areas of research would highlight any priority areas of research, and any areas lacking in resources. The bibliometric findings reflect the increased community awareness and prioritisation of domestic violence as an area for investigation, as the number of publications has continued to increase over time.

Of significance was the overall trend of an increasing number of publications per year, with this trend seemingly accelerating. This trend was consistent with those observed in previous reviews conducted by Jordan (2009) and Frank and colleagues (2010), indicating the growth of literature about domestic violence in Australia is consistent with worldwide trends. This growth was considered reflective of the increase in community awareness of domestic violence in Australia, and the Australian Government’s more recent prioritisation of this issue.

Several key documents concerning the state of domestic violence in Australia were also published just before or during 2011, including Mitchell’s (2011) parliamentary paper, Domestic Violence in Australia—An Overview of the Issues. This publication drew attention to the extent of domestic violence in Australia and may have sparked the reflected increase in publications. The increase in publications concerning domestic violence in Australia further highlights the need for bibliometric analysis, as it enables reflection upon features of existing literature and identifies gaps for future studies. Other key findings of the bibliometric analysis were that the overwhelming majority of retrieved documents were English articles; only four of the retrieved 669 publications were in a language other than English, and almost 80% of retrieved articles were published in Australia. These findings were unsurprising given “Australia” was included as a search term. Similarly, the top 10 ranking institutions and authors contributing to this body of research were Australian.

Professor Hegarty from the University of Melbourne was found to be particularly prolific, publishing more than double the number of documents of the next greatest contributing author. The University of Melbourne was the greatest contributing institution to this area of research, and each of the top 10 ranked institutions reflected the same trend of an increase in publications on this topic over time. As identified, the vast majority of publications within Australia were from Victoria, as four of the top ten contributing institutions were based in Victoria, including the University of Melbourne, Monash University, Deakin University, and La Trobe University. Queensland and New South Wales were the next biggest contributors, with two universities from each state recorded in the top 10 institutions. The remaining two institutions in the top ten were in South Australia. These findings appear reflective of the most prolific tertiary institutions in Australia, which are situated in the capital cities of the most densely populated states. However, they are not reflective of the rates of domestic violence across Australia, as a higher rate of domestic violence has been reported to occur in rural and remote areas (Mitchell, 2011).

The top 10 rankings of journals in this review reflected a diverse cross section of disciplines producing publications on the topic of domestic violence in Australia. These disciplines included health and medicine, interpersonal violence, psychology, criminology and law, and child wellbeing. Seven of the top ten ranked journals were Australian, and two of these were combined Australian and New Zealand journals. The remaining three were from the USA (two) and Canada (one). The interest of other countries in research into domestic violence in Australia is an area for further research, to determine whether articles published internationally are examining domestic violence on a global scale, or whether domestic violence in Australia is of interest to other countries.

## 5. Discussion

### 5.1. Key Findings

Existing research suggested risk factors for domestic violence are broad and varied [38,39,40] and there is a dearth of empirical evidence underlying current intervention programs in Australia [41,42]. Similarly, it is considered that practitioners and policy makers would benefit from an overview of features that contribute to the effectiveness of intervention for victims and perpetrators, and a greater understanding of factors that reduce intervention effectiveness. Information concerning domestic violence within Aboriginal and Torres Strait Islander populations is limited and data are largely estimated due to a lack of research and a pervasive trend of underreporting by Indigenous women [3,10,12,43,44]. It is therefore considered that additional research into patterns of domestic violence and underreporting within Indigenous populations is a further priority area for future research.

### 5.2. Future Recommendations

This study is the first bibliometric analysis of publications concerning domestic violence in Australia, and therefore provides valuable information concerning trends in publications. It is recommended this information is expanded in future studies. A priority area for future research is the analysis of publications that examine patterns of reporting and non-reporting of domestic violence, with a specific focus on barriers to reporting and factors that promote reporting. Additional priority areas for future research are examining the features and effectiveness of interventions for perpetrators and support services for victims of domestic violence.

### 5.3. Strengths

A key strength of this paper is that it is the first study to examine the bibliometric characteristics of publications regarding domestic violence in Australia. As a pioneering contribution to the literature, this publication provides valuable information about the state of research in this area for practitioners and policy makers, and provides guidance as to priority areas for future research. The bibliometric analysis is thorough and highlights valuable trends and patterns in the data. The inclusion of density visualisation mapping and ranking of information provides clear, visual representations of the data which afford the reader a snapshot of key features of this body of literature. Lastly, a high number of publications were extracted (673), ensuring the analysis was thorough and representative.

### 5.4. Limitations

In addition to strengths, this paper has several limitations. Articles were only extracted from Scopus. Therefore, any publications available solely through non-Scopus databases were not included. However, Scopus was selected due to its breadth of access to resources and features that facilitate bibliometric analysis that would not have been supported through other databases [19,24]. A further limitation was due to the nature of the bibliometric research: the information provided is reflective of broad trends and does not provide in-depth information concerning the quality or nature of the publications included. Therefore, conclusions cannot be drawn about specific patterns such as victimisation and intervention. This paper should, therefore, serve as a foundation for future research.

## 6. Conclusions

The current study sought to provide information regarding current research trends in domestic violence in Australia through analysis of bibliometric characteristics, with the goal of supporting practitioners, policy makers and researchers to better understand the research field. The study output provides clear information as to existing trends and information concerning research regarding domestic violence in Australia. The results support a trend of increased awareness and interest in domestic violence within Australia. Key additional findings included that the most prolific contributing institutions were located in more densely populated areas, despite a higher occurrence of domestic violence in rural and remote areas. Further, a diverse range of disciplines is connected to the study of domestic violence, including health and medicine, criminology and justice, and child wellbeing. This study also highlights key areas for future research, namely patterns of reporting of domestic violence, availability and effectiveness of interventions and services for perpetrators and victims of domestic violence, and examining patterns of domestic violence and reporting among Indigenous populations.

## Figures and Tables

**Figure 1 ijerph-19-04837-f001:**
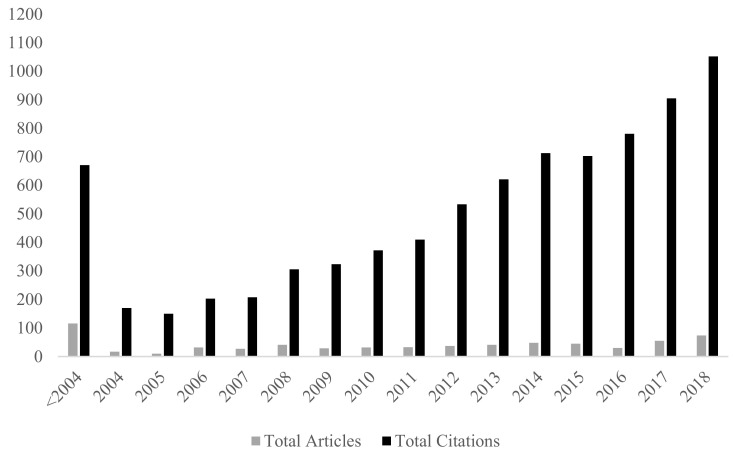
The number of publications and citations relating to search terms per year, between 1983 (grouped as prior to 2004) and 2018.

**Figure 2 ijerph-19-04837-f002:**
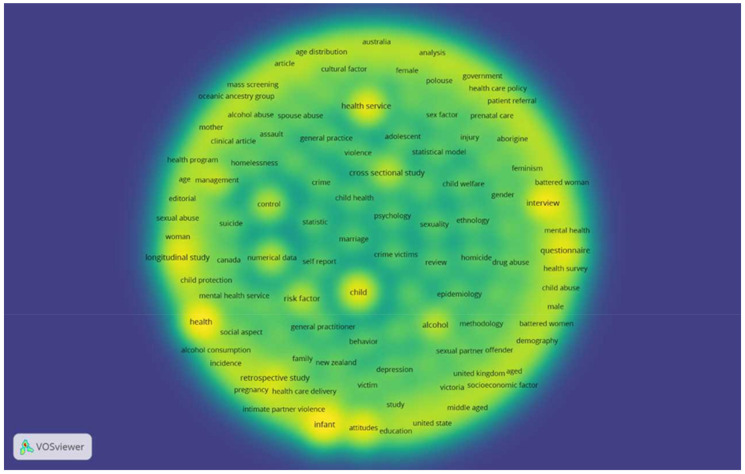
Density visualisation map of search terms encountered in the retrieved publications.

**Figure 3 ijerph-19-04837-f003:**
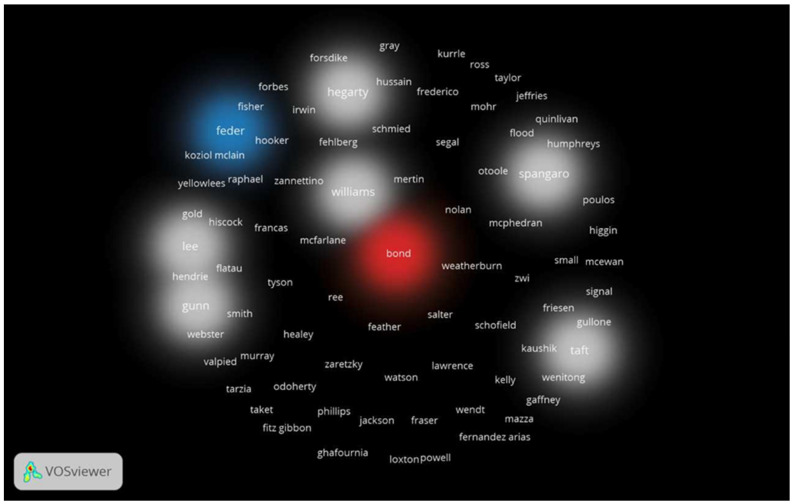
Density visualisation map of prolific authors encountered in the retrieved publications.

**Table 1 ijerph-19-04837-t001:** Types of documents.

Type of Document	Frequency (%)
Article	506 (75.2)
Review	57 (8.5)
Editorial	25 (3.7)
Article in Press	22 (3.3)
Book	17 (2.5)
Book Chapter	16 (2.4)
Note	11 (1.6)
Letter	8 (1.2)
Conference Paper	7 (1)
Short Survey	4 (0.6)
Total	669

**Table 2 ijerph-19-04837-t002:** Top 10 most frequently used search terms in articles extracted.

Rank	Search Term	Number of Occurrences
1	Human/s	701
2	Australia	409
3	Domestic violence	395
4	Female	319
5	Article	285
6	Adult	212
7	Male	187
8	Family Violence	135
9	Adolescent	120
10	Child	95
Total		2858

**Table 3 ijerph-19-04837-t003:** Ranked contributing countries in the number of domestic violence in Australia publications.

Rank	Country	Total Number of Documents (%)	Total Number of Citations (%)	Citation per Paper	*h*-Index
1	Australia	534 (79)	7183 (86.84)	13.45	43
2	UK	48 (7)	912 (11.03)	19	14
3	United States	47 (7)	674 (8.15)	14.34	13
4	Canada	22 (3.3)	498 (6.02)	22.64	10
5	New Zealand	20 (3)	256 (3.09)	11.64	8
6	Sweden	7 (1)	233 (2.82)	33.29	4
7	Switzerland	5 (0.75)	253 (3.06)	50.6	3
8	Chile	4 (0.6)	9 (2.13)	2.25	2
9	Belgium	3 (0.45)	176 (2.13)	58.67	1
10	Ireland	3 (0.45)	26 (0.31)	8.67	3

**Table 6 ijerph-19-04837-t006:** Top 10 institutions contributing to publications on domestic violence in Australia.

Rank	Institution	Country	Number of Published Articles (%)	Total Citations (%)	Citations per Article	*h*-Index
1	University of Melbourne	Australia	86 (12.78)	1644 (19.87)	19.12	21
2	Monash University	Australia	48 (7.13)	679 (8.21)	14.15	11
3	University of New South Wales	Australia	42 (6.24)	637 (7.70)	15.17	14
4	University of Queensland	Australia	39 (5.79)	883 (10.67)	22.64	15
5	La Trobe University	Australia	29 (4.31)	503 (6.08)	17.34	14
6	Deakin University	Australia	29 (4.31)	219 (2.65)	7.55	8
7	University of Sydney	Australia	29 (4.31)	332 (4.01)	11.45	10
8	Griffith University	Australia	25 (3.71)	212 (2.56)	8.48	7
9	Flinders University	Australia	22 (3.27)	217 (2.62)	9.86	9
10	University of South Australia	Australia	20 (2.97)	359 (4.34)	17.95	11

**Table 7 ijerph-19-04837-t007:** Top 10 journals publishing material on domestic violence in Australia.

Rank	Journal	Origin of Journal	Total Number of Articles (%)	Total Number of Citations (%)	SJR	SNIP	Cite Score	Impact Factor	*h*-Index	Citation per Article
1	*Medical Journal of Australia*	Australia	35 (5.20)	653 (7.89)	16.65	24.69	7.75	4.227	12	18.66
2	*Australian and New Zealand Journal of Public Health*	Australia and New Zealand	23 (3.42)	638 (7.71)	12.89	16.80	8.62	1.889	13	27.74
3	*Journal of Interpersonal Violence*	USA	16 (2.38)	247 (2.99)	19.14	24.91	13.12	2.443	7	15.44
4	*Violence Against Women*	USA	16 (2.38)	262 (3.17)	16.47	23.51	9.27	1.558	8	16.38
5	*Australian and New Zealand Journal of Criminology*	Australia and New Zealand	11 (1.63)	34 (0.41)	8.74	13.87	6.74	0.651	4	3.09
6	*Australian Journal of Primary Health*	Australia	11 (1.63)	37 (0.45)	4.86	6.50	6.58	1.152	3	3.36
7	*Contemporary Nurse*	Australia	11 (1.63)	66 (0.80)	5.41	6.36	6.17	0.673	5	6
8	*Psychiatry Psychology and Law*	Australia	11 (1.63)	61 (0.74)	5.52	8.52	4.04	0.787	5	5.55
9	*Australian Nursing Midwifery Journal*	Australia	10 (1.49)	0 (0)	2.12	0	0.1	0.070	0	0
10	*Child Abuse and Neglect*	Canada	10 (1.49)	325 (3.93)	24.84	29.55	17.85	2.899	7	32.5

**Table 8 ijerph-19-04837-t008:** Top 10 sources of funding for publications on domestic violence in Australia.

Rank	Sources of Funding	Number of Funded Articles
1	Australian Research Council (ARC)	24
2	National Health and Medical Research Council (NHMRC)	12
3	Criminology Research Advisory Council, Australian Institute of Criminology	5
4	University of Melbourne	5
5	University of New South Wales	4
6	Victorian Health Promotion Foundation	4
7	Australian Educational International, Australian Government	3
8	State Government of Victoria	3
9	U.S. Department of Justice (DOJ)	3
10	Centre for Mental Health and Wellbeing Research (CMHWR)	2

## Data Availability

Not applicable.

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
