# Peer review of "Bibliometric and Density Visualisation Mapping Analysis of Domestic Violence in Australia Research Output 1984–2019"

_ijerph, 2022, doi:10.3390/ijerph19084837_

Round 1

Reviewer 1 Report

The period under consideration is only until 2018, why is it not considered until 2021?

What is the difference between domestic or family violence?

There are no hypotheses raised in the work.

Author Hegarty is mentioned in many places, but citations are not seen, respectively, with other authors.

Many journals are mentioned in the tables, but no citations are seen.

Figures 2 and 3 are difficult to see and need to be redrawn.

The cited literature is quite old, some of it is not even reviewed.

There are no written limitations on the research.

Author Response

Reviewer 1:

The period under consideration is only until 2018, why is it not considered until 2021?

Response to reviewer 1: This review focuses on the research output from 1984 to 2019 (35 years), the later years will be included for analysis in our future studies. In addition, we have 35 years of data to analyses that consumed min 2.6 years.

What is the difference between domestic or family violence?

Response to reviewer 1: Although the terms “Domestic violence” and “Family violence” are used interchangeably. The key difference to be noted is that, “family violence” is an offence committed by a family member against another family member. On the other hand, “domestic violence” is when an individual commits an offence or abuse to an intimate partner. The difference lies in the relationship between the offender and the victim—not the crime’s nature.

There are no hypotheses raised in the work.

Response to reviewer 1: The purpose of this article is to consolidate available knowledge and to emphasise the need to develop research trends, according to the prevailing problems.

Kindly refer

Section 1.5:

. Rationale and objectives

As identified, domestic and family violence is recognised as a prevalent, pervasive issue nationwide, with long-lasting detrimental and potentially devastating consequences for those involved. The incidence of domestic violence is high in Australia, as an estimated 34 per cent of women have been a victim of domestic violence, and this number was estimated to be higher for Indigenous women [3]. Significant impacts of domestic violence include detrimental effects on health, social, and financial domains for victims, perpetrators, and children [3]. The financial cost of domestic violence was estimated to be 8.1 billion dollars in the 2002-2003 period, which was expected to rise to 9.9 billion dollars in 2021-2022 if intervention did not occur [3]. The Australian Government has identified responding to domestic violence as a priority, in identifying incidents, supporting victims, and providing appropriate interventions for perpetrators [2-4].

As awareness regarding domestic violence has increased, research and publications on the topic have also increased, however trends in these publications in Australia are yet to be explored through a bibliometric review. The analysis of bibliometric characteristics of domestic violence research in Australia allows the consolidation of knowledge on the available Australian scientific research evidence. This in turn could aid practitioners, policy makers, government, justice department, and researchers better understand the research field and learn more about current research trends, their impact, and scientific collaboration.

Therefore, the aim of this study was to analyse the Australian research publications on domestic and family violence using a bibliographic and density visualisation mapping analysis of articles indexed in Scopus.

Author Hegarty is mentioned in many places, but citations are not seen, respectively, with other authors. Many journals are mentioned in the tables, but no citations are seen.

Response to reviewer 1: Thanks for pointing it out. The table references have been citied and the same has been updated in the manuscript.

Figures 2 and 3 are difficult to see and need to be redrawn.

We have produced  those figures in a legible form when we submitted.

But the Editorial team squeezed it.

We have redrawn the Figures 2 and 3 as per recommendation.

Response to reviewer 1:

The cited literature is quite old, some of it is not even reviewed. There are no written limitations on the research.

Response to reviewer 1: The references are cited as per the inclusion of data used for analysis (1984-2019). The limitations have been added under the subsection “5.4. Limitations”.

Reviewer 2 Report

I have reviewed this article, that analyses the Australian research publications on domestic and family violence using a large number of articles indexed in Scopus.

The article is thoroughly researched and rendered with precision. It presents valuable information, aimed at developing and strengthening research trends.

The boundaries of the investigation are described perfectly. The text comes up with a thorough and correct methodology, since the main articles included in the bibliometric analysis are specifically detailed, and the text explains the inclusion and exclusion criteria.

The bibliometric analysis is really useful as it identifies gaps for future studies.

On the other hand, there are easy-to-read results of the analysis. The tables and figures are correct and efficient;  specifically the illuminating figures 2 and 3 provide an overview of the research field.

In addition, the discussion comes up with revealing ideas about the need to develop research trends, according to the problems of each area. The text highlights some needs, as those ones related to Indigenous population, but also clarifies that the conclusions cannot be drawn about specific patterns such as victimisation and intervention. These are honest and practical aspects that I want to highlight.

Nevertheless, in my opinion, the most remarkable weaknesses of this study are:

-An explanation about the reasons why some documents have been excluded (those categorized as abstracts, dissertation reports, case reports, case series…) is missing, especially because one of the purposes of this article is to consolidate available knowledge. I recommend an explanation in this regard, as other limits of the investigation have been justified, such as the exclusive use of Scopus instead of other sources.

 -Repetition of the same ideas (for example, the correlation between population density and publication density, or the need for more attention on domestic violence research in rural and remote areas) weakens the text. This problem is evident in the two final sections. Greater conciseness is desirable. Because of it, I would suggest that “Future recommendations” and “Conclusion” form a single text without repetitions. In this way, the text would be strengthened.

-References 11 and 12 should be reviewed.

Author Response

Reviewer 2:

Nevertheless, in my opinion, the most remarkable weaknesses of this study are:

-An explanation about the reasons why some documents have been excluded (those categorized as abstracts, dissertation reports, case reports, case series…) is missing, especially because one of the purposes of this article is to consolidate available knowledge. I recommend an explanation in this regard, as other limits of the investigation have been justified, such as the exclusive use of Scopus instead of other sources.

Response to reviewer 2:

The refences are cited as per the inclusion of data used for analysis (1984-2019). The limitations have been added under the subsection “5.4. Limitations”.

 -Repetition of the same ideas (for example, the correlation between population density and publication density, or the need for more attention on domestic violence research in rural and remote areas) weakens the text. This problem is evident in the two final sections. Greater conciseness is desirable. Because of it, I would suggest that “Future recommendations” and “Conclusion” form a single text without repetitions. In this way, the text would be strengthened.

-References 11 and 12 should be reviewed.

Response to reviewer 2: Thanks for your suggestion

Documents categorised as only abstracts, dissertation reports, case reports and case series have been excluded because they do not contain or disclose the complete data of the research and hence these reports cannot be used for analysis in our review.

 The last two sections have been rearranged and added under the respective subsections for more clarity for the readers.

The references have been reviewed and updated.

Reviewer 3 Report

I found this article very interesting, as violence of any kind must be analysed in all its aspects in order to be eradicated. Therefore, the issue under discussion is of great relevance and originality.
This is a very well written and easy to understand article.
The methodology is well explained, although, as will be noted later, more data could be provided. The results are very clear.
The authors have carried out a good discussion of results and presentation of conclusions.

Therefore, it seems to me that the article can be published, although I would like to make a series of observations:

  • At the end of the introduction there should be a paragraph setting out the structure of the rest of the article.
  • A section on literature review should be included.

More information is missing in the methodology section. As an example, Figure 1 in the article "Valls Martínez, M. D. C., Santos-Jaén, J. M., Amin, F. U., & Martín-Cervantes, P. A. (2021). Pensions, Ageing and Social Security Research: Literature Review and Global Trends. Mathematics9(24), 3258". This other article is also a good example: “León-gómez, A., Ruiz-palomo, D., Fernández-gámez, M. A., & García-revilla, M. R. (2021). Sustainable Tourism Development and Economic Growth : Bibliometric Review and Analysis. 1–20”.

Once these recommendations have been incorporated, the article can be published.

Author Response

Reviewer 3:

Therefore, it seems to me that the article can be published, although I would like to make a series of observations:

  • At the end of the introduction there should be a paragraph setting out the structure of the rest of the article.
  • A section on literature review should be included.

More information is missing in the methodology section. As an example, Figure 1 in the article "Valls Martínez, M. D. C., Santos-Jaén, J. M., Amin, F. U., & Martín-Cervantes, P. A. (2021). Pensions, Ageing and Social Security Research: Literature Review and Global Trends. Mathematics9(24), 3258". This other article is also a good example: “León-gómez, A., Ruiz-palomo, D., Fernández-gámez, M. A., & García-revilla, M. R. (2021). Sustainable Tourism Development and Economic Growth: Bibliometric Review and Analysis. 1–20”.

Once these recommendations have been incorporated, the article can be published.

Response to reviewer 3: We appreciate the time you have taken to read our work. A brief aim and objective of this study has been added under the subsection “1.5. Rationale and objectives” in the Introduction. Since we have mentioned or included relevant information with appropriate citations in each section, we did not provide a separate section for review of literature. Moreover, the entire review itself is a collection of available knowledge and analyzed data from previously published articles.

Round 2

Reviewer 2 Report

I think the text has improved enough. This article is an important contribution in the field. Thanks for your efforts. 

Reviewer 3 Report

After analysing the work done by the authors, I consider that this article can now be published.